# Sonodelivery in Skeletal Muscle: Current Approaches and Future Potential

**DOI:** 10.3390/bioengineering7030107

**Published:** 2020-09-09

**Authors:** Richard E. Decker, Zachary E. Lamantia, Todd S. Emrick, Marxa L. Figueiredo

**Affiliations:** 1Department of Basic Medical Sciences, Purdue University, 625 Harrison St., West Lafayette, IN 47907, USA; deckerre@purdue.edu (R.E.D.); zlamanti@purdue.edu (Z.E.L.); 2Department of Polymer Science & Engineering, University of Massachusetts, 120 Governors Drive, Amherst, MA 01003, USA; tsemrick@mail.pse.umass.edu

**Keywords:** sonoporation, sonodelivery, ultrasound, intramuscular gene delivery, gene therapy

## Abstract

There are currently multiple approaches to facilitate gene therapy via intramuscular gene delivery, such as electroporation, viral delivery, or direct DNA injection with or without polymeric carriers. Each of these methods has benefits, but each method also has shortcomings preventing it from being established as the ideal technique. A promising method, ultrasound-mediated gene delivery (or sonodelivery) is inexpensive, widely available, reusable, minimally invasive, and safe. Hurdles to utilizing sonodelivery include choosing from a large variety of conditions, which are often dependent on the equipment and/or research group, and moderate transfection efficiencies when compared to some other gene delivery methods. In this review, we provide a comprehensive look at the breadth of sonodelivery techniques for intramuscular gene delivery and suggest future directions for this continuously evolving field.

## 1. Introduction

Gene therapy shows increasing promise for the treatment of a wide range of diseases, from cancer to musculoskeletal and immunological disorders [1]. Although the basic premise of gene therapy is relatively simple on the surface—to introduce a gene or genes into a patient’s cells/tissue that will allow the body to produce a therapeutic protein or facilitate a genetic modification—every step of the process holds its own challenges. One of the major challenges associated with making gene therapy widely (and safely) accessible is that of gene delivery in vivo. There are currently many methods for delivering DNA in vitro at high efficiencies, including electroporation, lipid-based transfection, and viral delivery. Each of these methods presents potential challenges for clinical translation, including the cost and time of production, the invasiveness of the technique, the efficiency of delivery, and/or safety concerns related to an immune response or unintended genomic integration of the therapeutic gene. The development of a gene delivery technique that is inexpensive, widely available, reusable, minimally invasive, and safe will catalyze progress in gene therapy. One promising technique that meets all of these criteria has been under investigation for many years but has not yet reached widespread use: ultrasound-mediated or sonoporation gene delivery, herein referred to as sonodelivery [2,3]. As its name suggests, this approach involves using ultrasound to deliver non-viral vectors encoding therapeutic proteins into skeletal muscle, allowing the body to then produce its own therapeutic. Despite the work that has been done to explore the practicality of ultrasound-mediated intramuscular gene delivery, there is, to our knowledge, no comprehensive review of the most common approaches and technical conditions used for this promising technology. This review aims to fill that void and to discuss the future work needed for and the potential of establishing sonodelivery as a key player in the gene therapy field.

## 2. Methods for Cellular and Intramuscular Gene Delivery

There are many established methodologies for delivering plasmid DNA (pDNA) to muscles, each with its own advantages and disadvantages (Figure 1). To highlight these and provide context for the importance of sonodelivery, we will provide a brief background on four of the most common pDNA delivery approaches (excluding sonodelivery).

### 2.1. Direct Injection of Naked DNA 

The oldest and most basic method of intramuscular DNA delivery is injection of naked DNA directly into the target muscle. This method yields a low immunogenic response in vivo and is both widely available and inexpensive [4]. Plasmids are low-cost and low maintenance and intramuscular injections are straightforward (though not necessarily easy) to administer, making this method one of the simplest to apply. However, it suffers from a very low efficiency [5], which has prevented direct injection of naked DNA from becoming a viable technique. It has been reported that swelling the muscles before injection (e.g., pretreating with a hypertonic solution) improves expression of proteins for which the transfected genes code [4,6,7]. Similarly, introducing myotoxic agents to the target muscle a few days prior to DNA injection has been shown to improve protein expression, but it is not yet clear whether this improvement is due to improved uptake of naked DNA or improved gene expression following uptake [4,8]. With all intramuscular gene delivery, injection technique is crucial [9]. A properly localized injection at optimal depth is essential, but is increasingly challenging as the size of the in vivo model increases [4]. Injections are therefore difficult to standardize, as optimal techniques require training and extensive practice. One method that has been used to increase injection accuracy when delivering a variety of payloads is the utilization of imaging ultrasound to guide the injection [10,11,12,13], although this too requires expertise and extensive practice. Another complication associated with increased test animal size is that of decreased gene uptake/expression. This may be due to increased levels of connective tissue in larger animals, which either acts to disperse the naked DNA or as a barrier to the DNA, preventing it from entering muscle cells [14]. Despite extensive research involving direct injection of naked DNA, DNA uptake and protein expression levels remain low. Thus, efforts to improve the efficiency of DNA delivery, including both biochemical and physical approaches, have led to strategies utilizing polymer and nanoparticle coupling, viral vectors, and electroporation, among others.

### 2.2. Polymers and Nanoparticles

Polymer and/or nanoparticle conjugation to DNA involves incubating DNA with the polymer/nanoparticle sample prior to delivery to improve transfection efficiency. These particles, if engineered with the appropriate properties, generally do not compromise cell viability or cause significant immunogenicity [15]. Interestingly, several aspects of the polymer, such as molecular weight, stability, size, charge, and structure, greatly affect the efficacy of the DNA binding and subsequent delivery into cells [15]. The ability to change these factors through polymer engineering makes this method highly adaptable; for example, polymers of a certain molecular weight might work well for intramuscular gene delivery but be ineffectual when applied to hepatocytes. It is also of note that polymers are often easy and inexpensive to synthesize [15]. Cationic polymers form so-called “polyplexes” (polymer–DNA complexes) with DNA, by binding to negatively charged phosphate groups in DNA, which facilitates the introduction of the DNA into the cell [16]. Despite their relative ease of use and production, these formulations are still inferior to alternative gene delivery methods in terms of efficiency (e.g., viral vectors), although they have been shown to approximate the efficiency of viral vectors when used in combination with physical methods of gene delivery [17,18,19]. Additionally, both degradable and non-degradable polymers have drawbacks that, to date, have complicated clinical applications. Degradable polymers may lead to accumulation of toxic metabolites and adverse reactions to treatment [20], whereas non-degradable polymers may aggregate during circulation and present a risk of clotting [21,22].

### 2.3. Viral Vectors

Viral vectors boast some of the highest gene delivery efficiencies of any of the methods discussed herein [10,23]. Multiple types of viruses are used in transfection, with adeno-associated viruses (AAVs) among the most common, especially for delivering genetic material into muscle cells. Although it is a significant challenge to produce good manufacturing practice (GMP) levels of virus at high enough titers for clinical trials, localized and systemic gene transfer methods utilizing AAVs have been developed, each with its own advantages and disadvantages [19,24]. Systemic viral delivery (i.e., intravenous injection) poses great risk to patients due to potential immune response and off-target delivery [13]. Localized delivery (i.e., direct intramuscular injection) tends to be much safer than systemic delivery because it induces a lower level of immune response and has decreased off-target delivery, but it is often less efficient for transducing large muscle masses due to the inability of the viral vector to disperse throughout the muscle tissue. In tumors, it is possible to use oncolytic viruses that can spread throughout the mass, but this would likely not be a safe approach for muscle. With appropriate targeting, the risk of infecting surrounding healthy myocytes can be reduced, though not eliminated entirely [25]. An additional risk of viral gene delivery is insertional mutagenesis. Through the process of genomic integration, additional base pairs are introduced to the genome [26]; these unintentional additions can be oncogenic and/or cause a great number of genetic defects. In summary, there are a number of major challenges associated with the most commonly used viral vectors, from the high risk for immunogenicity to the risk of (unintended) insertional mutagenesis to the high cost of production [19,23,27].

### 2.4. Electroporation

Due to the immunogenicity risks associated with biochemical methods, physical approaches have recently been investigated more extensively. One physical delivery method that is well integrated in the gene delivery field is electroporation. There are a range of electroporation devices and techniques, with many different products on the market to fit specific applications in vivo, in vitro, in situ, and ex vivo. Electroporation tends to produce a high transfection efficiency in all applications [28]. Its mechanism of action lies in the permeabilization of the cell and nuclear membranes using electricity [29]. The electrical current (“electro-”) creates transient pores (“-poration”) in the cell membrane through which genetic material can enter the cytoplasm [30]. The DNA can then move into the nucleus through similar pores in the nuclear membrane. This method can be applied to any portion of the body and is a relatively rapid process. One of the main challenges involved with electroporation is the invasiveness of the technique in vivo; electrodes are typically inserted into the subject [13,31]. Another major challenge with electroporation is the resultant low cell viability in vivo [32]. Gaps in the cell membrane can cause both an inflammatory response and necrosis of muscle fibers, which can lead to muscle damage. Although there are many different settings that can be adjusted to dramatically alter the uptake of DNA and/or viability of cells, electroporation still tends toward low viability [33]. Finally, electroporation is costly when compared to most other methods of physical gene delivery.

## 3. Ultrasound-Mediated Gene Delivery in Skeletal Muscle

Although each of the above-mentioned techniques has advantages and are helping to further the progress of gene therapy, their disadvantages are significant. One promising solution for delivering non-viral vectors to muscles for therapeutic protein production is ultrasound-mediated intramuscular gene delivery (sonodelivery), with advantages pertaining to maximizing safety, accessibility, and efficiency, while minimizing cost and invasiveness. 

Sonodelivery is intramuscular gene delivery augmented with ultrasound to facilitate increased uptake of DNA. Although some of the general mechanisms will be discussed herein, a description of the basic methodology used for sonoporation will not be provided. We would encourage anyone interested in a thorough description of in vitro and in vivo sonoporation methodologies to refer to the previously published works available, including [13,34,35], among others.

### 3.1. Benefits of Ultrasound

Patient safety and therapeutic expense are two major factors to consider in any treatment type. Consequently, ultrasound-based strategies hold great promise for gene delivery or other therapeutics, due to their low cost and clinically established safety [13]. Ultrasound has been used on patients since the late 1940s and has been considered safe for pregnant women since the 1950s [36]. More specifically, the levels of ultrasound required for sonodelivery are also associated with low cytotoxicity levels and minimal tissue damage in vitro and in vivo [2,37]. Although sonodelivery is typically performed with specialized equipment, standard clinical ultrasound machines can be modified to produce the necessary output [10,18,38,39]. Standard ultrasound equipment is widely available and reusable indefinitely, making ultrasound a very affordable means of physical gene delivery.

An important consideration for a gene delivery system is its degree of invasiveness. As ultrasound waves can penetrate deep tissue even when applied epicutaneously, it is a very minimally invasive technique [3,40]. This combination of affordability, minimal invasiveness, and high level of safety makes ultrasound an enticing option for clinical gene delivery.

### 3.2. Sonoporation In Vitro

Generally, biological methods are tested in vitro prior to performing in vivo studies. This has been the case with sonoporation, with groups testing the efficacy of sonodelivery in various cell types [34,41,42,43]. Prior studies indicate that the conditions for in vitro sonodelivery tend to vary widely, from the settings of the sonoporator (Table 1) to the way cells are prepared for gene delivery (e.g., in suspension or seeded in culture plates). This wide variance tends to lead to variable transfection efficiencies. While it is common to experience problems when translating from in vitro to in vivo, this problem is particularly pronounced for sonodelivery, making it very challenging to effectively optimize conditions for intramuscular gene delivery using in vitro testing data [34,39,41,42]. There are likely many factors leading to this, such as increased difficulty in creating pores in cells in vivo [41], the rigid plastics that are used for cell culture causing ultrasound beam distortion issues [42] and the creation of standing waves [39], differences in DNA accessibility in culture versus in the natural muscle environment, and effectiveness of cell repair mechanisms in a monoculture. Other factors may also include the in vivo structure being more “protective” of the cells or the body’s ability to recruit inflammatory cells to the muscle following sonoporation.

We expect sonoporation to eventually become an inexpensive, reusable option for transfecting cells, especially following optimization and commercialization of specific tools designed to augment sonoporation in vitro. A significant challenge is the lack of well-defined sonodelivery conditions employed in vitro across laboratories (Table 1). This, combined with the lack of in vitro–in vivo correlation, establishes the necessity of further sonodelivery research, especially in vivo. Particularly notable is that, in our experience, in vivo sonoporation is more effective than in vitro sonoporation. As such, we will focus this review on in vivo sonodelivery work, as it appears to share more similarities across laboratories on how skeletal muscle sonoporation is approached and performed and we anticipate that it will have the greatest impact on the field of gene therapy.

### 3.3. Skeletal Muscle as a Therapeutic “Factory”

Although sonodelivery could be used to deliver genes in vivo for short-term treatment therapies such as targeted gene editing systems (e.g., CRISPR/Cas9), to deliver a therapeutic protein or drug directly to the site of interest [48,49], or to deliver viral vectors [50] or siRNA [51,52], this review will focus primarily on the delivery of plasmid DNA for long-term protein production. 

Unlike injecting a dose of a drug that will be metabolized relatively quickly, genetic material must remain in the body long enough to allow for protein production and for the therapeutic protein to reach levels sufficient to treat a patient’s condition. Their low turnover, high metabolic rate, and ability to secrete cytokines and proteins (myokines) make skeletal muscle cells ideal candidates for protein production, particularly secreted proteins with therapeutic activity. Skeletal muscle’s inherent ability to repair itself using progenitor cells (i.e., satellite cells) could also provide a potential healing mechanism for any localized toxicity or inflammation induced during gene delivery [29,53,54]. The concept of gene therapy, to use a patient’s own tissue to create a therapeutic by delivering genes coding for therapeutic proteins, relies heavily on the tissue’s ability to produce these proteins over time. Skeletal muscle, which makes up approximately 30% of and provides structure and stability throughout the entire adult human body [55], is a promising target for therapeutic protein production because its cells tend to live longer than other cell types. A muscle can effectively become a “factory” within the body, continuously producing therapeutic proteins for months at a time [56,57] if the vector enables long-term expression. The therapeutic proteins produced at the factory can then be delivered systemically [43,55,58], allowing for long-term treatment following a single dose of intramuscular gene delivery [59]. Fusing specific motifs to the protein can further facilitate targeted delivery, for example, thus increasing therapeutic effectiveness and safety by decreasing off-target accumulation [60].

### 3.4. Sonoporation in Skeletal Muscle

Sonoporation has been proven effective for delivering pDNA into cells and tissues, especially when used in tandem with microbubbles (MBs) [19,38,43,61,62]. In studies using firefly luciferase as a reporter gene, pDNA, supplemented with MBs, was delivered successfully to muscles in vivo and the luciferase protein was expressed at high levels (nearly 100-fold higher than the non-ultrasound control) for prolonged periods of time (Figure 2) [10]. To our knowledge, studies have only shown localized luciferase activity in muscle cells at or near the injection site, with no indications of off-target delivery, accumulation, or expression [39].

One of the key components for successful sonoporation in muscle in vivo is the addition of MBs [10,19,38,43,62]. MBs are common echo-contrast agents used in medical procedures in humans. They typically consist of an inert gaseous bubble encapsulated by a biocompatible material, with lipids being the most common interfacial encapsulant [13,63]. For sonodelivery, MBs can either be fabricated with a protein or gene of interest loaded within the bubble or “empty” bubbles can be complexed with the payload prior to injection [64]. Examples of microbubbles used for sonoporation include SonoVue^TM^ (Bracco), which has a phospholipid shell surrounding sulfur hexafluoride, and Optison (GE Healthcare), which consists of albumin-encapsulated perfluoropropane [65,66]. It is not fully understood how sonodelivery works at the cellular level, but it is generally believed that ultrasound energy produces transient micropores in the cell membrane via cavitation caused by oscillation of small bubbles present in the ultrasound media [2,43,45,55,67,68]. It is further believed, and empirically supported, that the addition of MBs significantly increases the cavitation facilitated by ultrasound alone (Figure 3) [13,43]. It has also been suggested that MBs create a microjet stream when they rupture, which forces the bound (or encapsulated) pDNA (or protein) into the cells through the pores created by the ultrasound-induced cavitation [44,64]. Regardless of the mode of action, it has been shown repeatedly that the addition of MBs significantly increases payload delivery into skeletal muscle as well as other types of tissue [10,19,55,61,69,70,71,72].

While combining ultrasound and MBs effectively delivers luciferase DNA into muscle, delivery of other types of DNA is problematic, possibly due to inefficient DNA uptake into the nucleus from the cytoplasm. While this would not be problematic for protein delivery, DNA must be taken into the nucleus for protein production to occur. One way to overcome this is by incorporating polymers with a nuclear localization sequence (NLS). A previous study conducted by our group showed that sonodelivery of DNA combined with MBs and an NLS-tagged polymer did indeed increase the efficiency of plasmid uptake and subsequent protein production by as much as six times compared to the non-US control [17] (Figure 4). On the other hand, some untargeted polymers, such as P85 Pluronic block copolymer, combined with MBs and ultrasound, showed a ~2-fold increase in pDNA delivery when compared to the non-ultrasound control [18]. Although further work is needed to determine the best polymer properties and ratio to MBs and pDNA and to determine the ideal ultrasound settings to deliver the polymer/MBs/pDNA mixture, this approach holds great promise for improving the efficacy of intramuscular sonodelivery. NLS polymers with variable compositions to balance hydrophilicity and charge are currently under development with the objective of further enhancing sonodelivery in muscle.

### 3.5. Effects of Sonoporation on Skeletal Muscle

Studies focused on how skeletal muscle is affected by sonoporation further demonstrate that it is one of the safest methods for gene delivery in vivo. For example, Chen et al. showed that muscle damage was comparably minimal following MB-free DNA injection with or without polymer and with or without ultrasound. When MBs were combined with ultrasound, there was a statistically significant increase in muscle damage compared to the control (Figure 5 and Figure 6). The authors noted that the damage was always localized around the injection site even when gene expression was observed distally [18]. In a similar study conducted by our group, muscles that had been injected with a DNA/MBs/polymer mixture and then subjected to ultrasound showed no significant damage at any of the time points evaluated during the study: 7 days, 23 days, or 12 months (Figure 7) [59]. Two other studies, conducted by Shapiro et al. and Wang et al., support our findings, with results showing no significant damage to muscles following treatment with MBs, DNA, and ultrasound [10,38]. Interestingly, a study conducted by Lu et al. yielded results contradictory to those of Chen et al., in that muscle damage was decreased in mice treated with DNA, MBs, and ultrasound compared to mice treated with DNA only [19]. Overall, these results are promising, especially in comparison to electroporation, which consistently induces muscle damage [29,73]. If we also consider the remarkable ability of skeletal muscle to repair itself from minor damage by employing progenitor (satellite) cells [53,54], the damage caused by sonoporation may not have a significant adverse impact on muscle fibers.

### 3.6. Variability in Sonoporation Conditions

Another factor that may lead to discrepancies between luciferase delivery and delivery of other types of DNA is variability in sonoporation conditions. Other variables may include the type of luciferase gene used (firefly, Renilla, enhanced firefly, etc.), the promoter driving transcription of the gene and other vector-related properties, the composition of the MB stabilizer (polymer, lipid, etc.), the concentration of MBs, and whether or not polymers are used in the nanoplex formulation. As was shown in Table 1, most research groups tend to use different sonoporation conditions in vitro. Table 2 shows that this is further accentuated for in vivo work. There are also a variety of different sonoporators in use, with many being custom modified ultrasound instruments. Although there are some commonalities among the different techniques, there is no consensus on an “ideal” set of conditions. Conditions used are largely dependent on each research group’s empirically derived preferences and the equipment used for the experiment. Although each research group has had success with their chosen sonoporation method, these differences could contribute to the disparity of results in the efficiency of sonodelivery and subsequent protein production. Developing a standardized protocol could help to bring defined sonoporation-mediated gene therapy conditions to the clinical stage. On the other hand, the success of sonodelivery under a variety of conditions highlights its potential application in a variety of clinical scenarios. For example, patients of different demographic groups (especially age) will likely have different levels of muscle density, so the ability to adapt the ultrasound conditions for each patient could increase the reach of sonodelivery.

## 4. Future Directions

The most pressing future direction for sonodelivery is that of standardization. With so many different sonoporators and sonoporation conditions, even for the same target tissue or cell type, and with such disparity in the results of the gene delivery, it is difficult to determine a standardized treatment method. Developing a standard unit of measure could help to expedite this process. Sonoporation conditions are typically reported by duty cycle, frequency, time, and intensity. Many reports also include probe size and some reports include other conditions, ranging from wave type to power output. Unfortunately, with the differences between instruments, the conditions that are typically reported can lead to a variable power output, meaning that the delivery power is likely not the same, even if the conditions are very similar. Although more optimization clearly is needed to find the average optimal conditions for delivering DNA into muscle using sonoporation, establishing a standard method for reporting sonoporation conditions, such as the inclusion of the power output, which can be calculated for each set of conditions, may help to ameliorate some of the disparities in results.

An interesting future direction for the sonodelivery field would include more work on combining sonoporation with other approaches that might increase the efficacy of gene delivery. Some work has already been done along this vein, including polymer-mediated sonodelivery [17,18,59], adjustments to cell membrane permeability by adding enzymes, such as hyaluronidase [80], and by adding drugs such as lidocaine and adjusting temperature [41], and electro-sonoporation [58,81,82]. Electro-sonoporation is a particularly interesting procedure in which electroporation and sonoporation are applied simultaneously to facilitate cell membrane pore formation. In a proof of concept investigation, Longsine-Parker et al. found that cell viability was 97.3% and membrane pore formation efficiency was 95.6% in mammalian cells after application [82]. These promising results were further explored with in vitro and in vivo studies that found that electro-sonoporation performed significantly better in both transfection efficiency and cell viability compared to electroporation alone [58,81]. While more work is needed to confirm these results and elaborate on the effectiveness of electro-sonoporation, it appears to be a highly promising technique. While all these possible adaptations would add more conditions to consider in the standardization procedure, they could greatly enhance the efficacy of sonodelivery for gene therapy.

Another direction of high importance and interest for intramuscular sonoporation will be the delivery of therapeutic genes, as opposed to reporter genes (i.e., luciferase). Although sonoporation has been used to deliver therapeutic genes intratumorally [62,76], the majority of the intramuscular sonodelivery completed to date has utilized reporter genes to test sonoporation conditions and efficiency. It could be argued that testing the delivery of therapeutic genes via sonoporation is of greater importance than standardization, as having a standardized method will only be useful when sufficient levels of therapeutic expression are observed. To help mitigate the potential for non-standard conditions to be a confounding factor of a lack of therapeutic expression, this work could be conducted in tandem with the above-mentioned standardization work. Of particular interest would be the delivery of therapeutic proteins tagged or fused with a reporter gene, such as luciferase, which would allow for simultaneous testing of therapeutic expression and optimization of sonodelivery conditions.

Further improvements for intramuscular sonodelivery and gene therapy might include the incorporation of muscle-specific refinements of payload, such as microbubbles designed to target skeletal muscle or the extracellular matrix (ECM) or vectors containing skeletal muscle-specific promoters to increase the efficiency of transgene expression [83]. In light of all these considerations, it can easily be seen that there is additional optimization needed for these approaches and there would be great benefit to the scientific community in implementing more extensive testing of therapeutic delivery to skeletal muscle using sonoporation.

## 5. Conclusions

As gene therapy becomes more popular as a viable therapeutic strategy, an affordable, non-invasive, non-viral gene delivery system will help to increase its availability and safety. Ultrasound-mediated gene delivery can fill this niche and, with continued study and proper standardization, sonodelivery has the potential to become the predominant technique for intramuscular gene delivery.

## Figures and Tables

**Figure 1 bioengineering-07-00107-f001:**
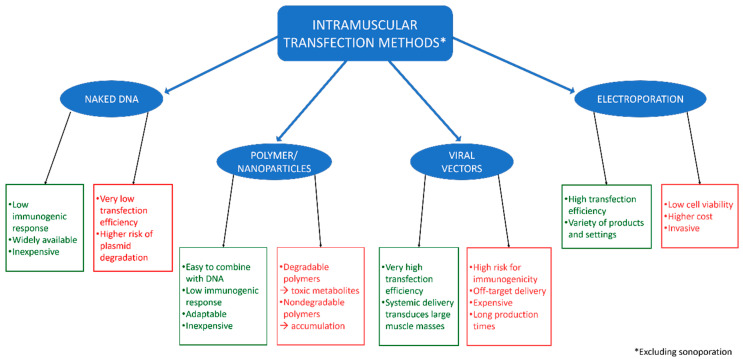
Common approaches for delivery of plasmid DNA into skeletal muscle and their associated advantages (green) and disadvantages (red).

**Figure 2 bioengineering-07-00107-f002:**
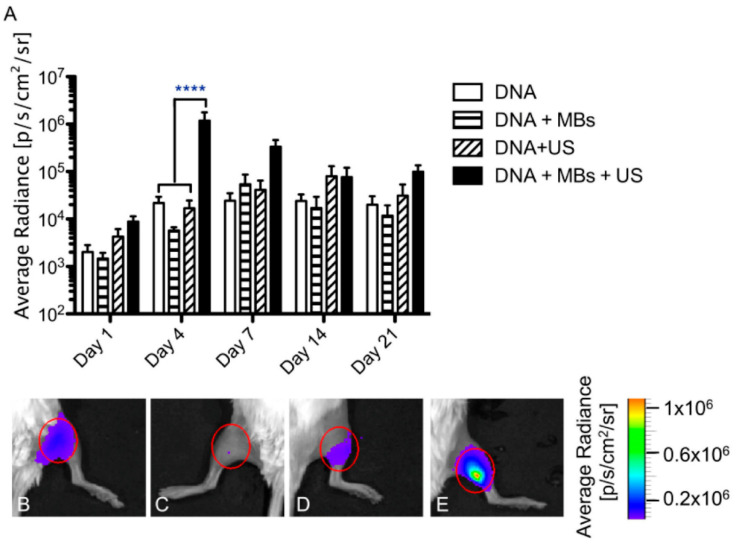
Luciferase 2 (Luc2) expression time course following sonoporation in vivo. (**A**). Luciferase expression profile in mice treated with microbubble-enhanced sonoporation of Luc2 (an enhanced form of Luciferase) in vivo. Mice were injected once with 50 μg plasmid DNA (DNA and DNA + ultrasound (US) groups), or 50 μg plasmid DNA premixed with 5 × 10^5^ microbubbles (DNA + microbubbles (MBs) and DNA + MBs + US groups) and then treated with an acoustic pressure of 200 kPa for 2 min (DNA + US and DNA + MBs + US group). The treatment effect was monitored using bioluminescence imaging for 21 days after treatment. ****, *p* < 0.05 relative to control. (**B–E**). Representative images obtained in each group (DNA, DNA + MBs, DNA + US, and DNA + MBs + US, respectively) on Day 4 with the region of interest indicated by a red oval. Color bar, p/s/cm^2^/sr. (Reprinted from Shapiro et al. [10] with permission.)

**Figure 3 bioengineering-07-00107-f003:**
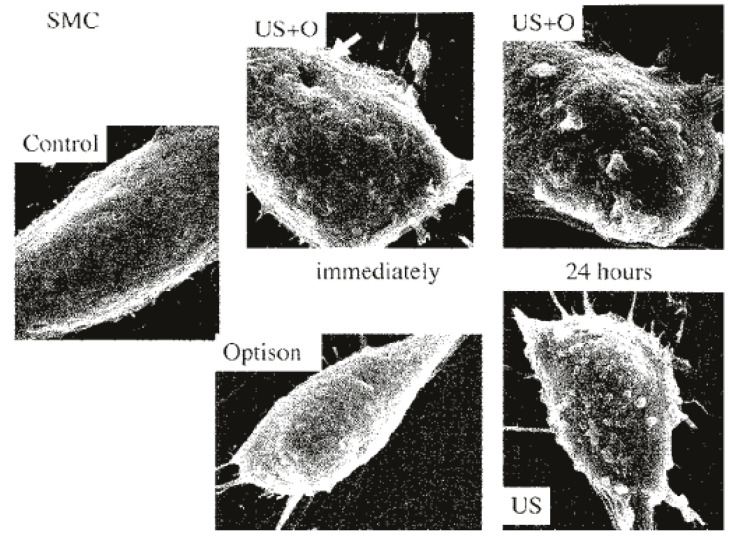
Electron microscope images of human skeletal muscles transfected with Optison MBs and pDNA with or without US. Micropores can be seen immediately following transfection (white arrow). (Reprinted from Taniyama et al. [43] with permission.)

**Figure 4 bioengineering-07-00107-f004:**
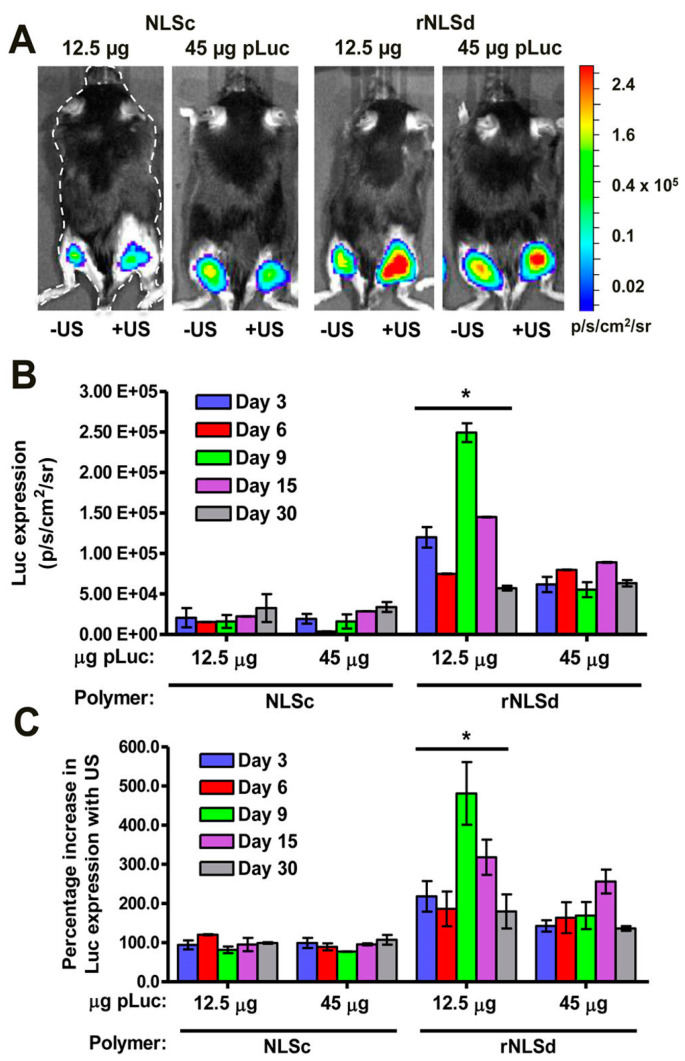
Intramuscular ultrasound-mediated gene delivery or “sonodelivery” in mice by NLSc- and rNLSd-based polyplexes. (**A**) Representative bioluminescence images of mice transfected with a luciferase reporter plasmid after a 5 min acquisition time using a Xenogen IVIS100 CCD camera (9 days after sonoporation). The right hind legs represent protein expression resulting from intramuscular ultrasound-mediated delivery; the left hind legs represent intramuscular delivery in the absence of ultrasound. The color bar indicates the luminescence intensity in photons per second per square centimeter per steradian (p/s/cm^2^/sr). (**B**) Luciferase expression from right hind legs. (**C**) Percentage change in luciferase expression upon application of an ultrasound stimulus (the ratio of the expression from the right hind leg to that from the left hind leg). Bars indicate ± the standard error of the mean (error bars) with * *p* < 0.05 for the rNLSd 12.5 μg dose as compared to all other groups. (Reprinted with permission from Parelkar et al. [17]. Copyright (2014) American Chemical Society.)

**Figure 5 bioengineering-07-00107-f005:**
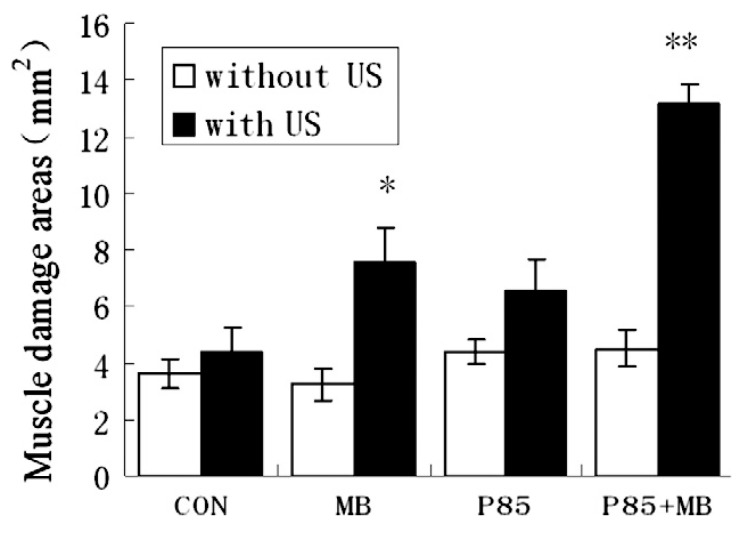
Sizes of the regions of damaged muscles. Both * and ** indicate comparisons between groups and saline control. * indicates *p* < 0.05 and ** *p* < 0.01. Data are presented as mean ± s.d. (two-tail Student’s t-test, Microsoft Excel 2003). (Reprinted from Chen et al. [18] with permission.)

**Figure 6 bioengineering-07-00107-f006:**
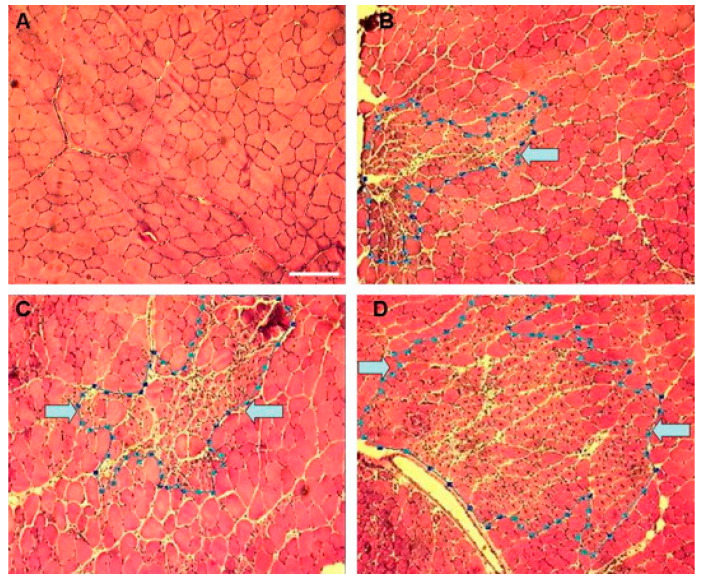
Images captured by 20× objective show damaged muscles after injection of transfection solution with or without ultrasound irradiation. Broken lines and arrows indicate the regions of injured muscles. Panel (**A**): no injection; panel (**B**): injection of transfection media containing DNA alone; panel (**C**): injection of transfection media containing DNA + 10% Optison; panel (**D**): injection of transfection media containing DNA + 10% Optison followed by US. The white scale bar indicates 100 μm. (Reprinted from Chen et al. [18] with permission.)

**Figure 7 bioengineering-07-00107-f007:**
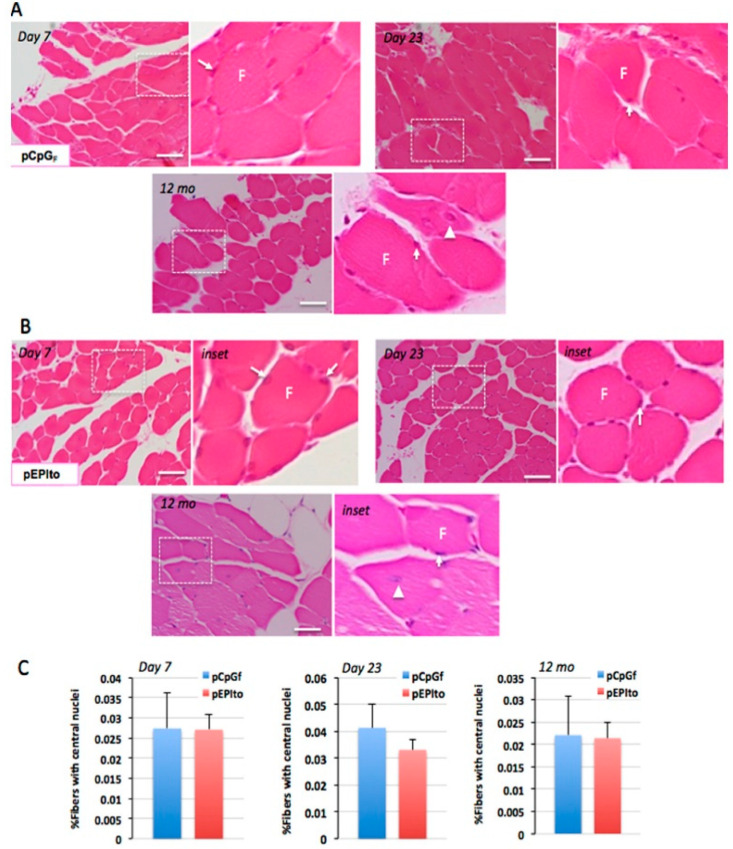
Analysis of muscle damage following gene delivery. Histological analyses at 7 days, 23 days, and 12 months, indicate neither plasmid, pCpGF (**A**) nor pEPIto (**B**), produced any significant signs of damage, as shown by H&E staining and visualization of skeletal muscle at 200× magnification. Inset: selected areas at higher magnification to show nuclear positioning in muscle fibers. Scale bar represents 50 μm. Arrows: normal positioning of nuclei at the periphery of muscle fibers, F; arrowheads: occasional atypical nuclei positioned centrally within fibers. (**C**) Analyses of percentage of muscle fibers displaying atypical central nuclei show no significant differences (*p* > 0.3) for either plasmid group for all three time points. (Reprinted from Figueiredo Neto et al. [59] with permission.)

**Table 1 bioengineering-07-00107-t001:** Examples of common in vitro sonoporation conditions. Conditions listed are representative of the preferred conditions for pDNA delivery in the referenced experiment, not representative of all conditions tested. PRF = pulse repetition frequency; “-“ = not reported; * = unpublished data; ** = therapeutic instrument.

Sonoporator	Frequency (MHz)	Burst (W/cm^2^)	PRF (Hz)	Duty Cycle (%)	Duration (s)	Cell Type	Reference
KTAC-4000	1.015	0.7	-	50	360	C2C12	Our Group *
KTAC-4000	2	2.5	2	50	10	COS-7	[44]
Sonidel SP100	1	2	100	60	450	HeLa	[42]
Sonidel SP100	1	4	-	60	450	C2C12	Our Group *
Panametrics	2.25	-	100	20	10	MAT B III	[45]
ES-1 Ultrasonic Generator	1	3.6	-	-	20	PC3	[41]
“Dedicated Continuous Wave System”	1	0.75	-	-	30	VSMC, HUVEC	[39]
Mark 3, EMS Limited **	1	0.8	100	20	20	H2K Myoblast	[46]
UltraMax **	1	2	100	30	1800	BHK, LNCaP, BCE	[47]

**Table 2 bioengineering-07-00107-t002:** Examples of optimized in vivo sonoporation conditions. The conditions listed are representative of the preferred conditions for pDNA delivery in the referenced experiment; they are not representative of all conditions tested. PRF = pulse repetition frequency; “-“ = not reported; * = unpublished data; + = not listed in work, but based on work carried out by same reference group.

Sonoporator	Frequency (MHz)	Burst (W/cm^2^)	PRF (Hz)	Duty Cycle (%)	Duration (s)	Reference
**Dedicated Sonoporators**						
BFC Applications Probe + WF1946A Frequency Synthesizer	1	3	1000	20	60	[74]
KTAC-4000	1.015	3	-	20	60	Our Group *
Sonidel SP 100	1	2	-	50	180	Our Group *
Sonidel SP 100	1	1.9	-	25	360	[42]
Sonidel SP 100	1	2	-	25	180	[75]
Sonigene	1	3	-	20	60	[76]
Sonigene	1	2	-	20	60	[17]
Sonitron 2000	1	0.4	200	20	1200	[55,77]
Sonitron 2000	1	5	-	50	600	[78]
Sonitron 2000	1	4	-	50	300 (60 * 5)	[79]
Sonopore 3000	2	2.5	2	50	60	[51]
**Therapeutic Instruments**						
Mark 3, EMS Limited	1	1	-	20	120	[18]
Mark 3, EMS Limited+	1	3	100	20	60	[19]
Mark 3, EMS Limited	1	2	100	20	30	[38]
Modified Siemens Antares	1.4	-	540	-	120	[10]
Ultax UX-301+	1	2.5	-	-	60	[43]
UltraMax	1	1.5	-	30	120	[80]
System V, GE Vingmed	1.7	Mechanical Index = 1.7			180	[39]

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
