# Peer review of "Sonodelivery in Skeletal Muscle: Current Approaches and Future Potential"

_bioengineering, 2020, doi:10.3390/bioengineering7030107_

Round 1

Reviewer 1 Report

Decker et al. presented the pros and cons of sonoporation as a gene delivery method.

Sonoporation as a method of intramuscular gene delivery is a technique of great importance. However, it seems that the theme is described very generally. First of all, I do not see the need to briefly describe other methods as this information does not bring anything significant to the main issue, which is sonoporation. If other methods of gene delivery are described, the purpose of the manuscript should be changed to present a critical view of the various gene delivery techniques. Moreover, the manuscript does not bring new information that would help researchers to choose this method for their experiments. The authors have experience in performing sonoparations both in vitro and in vivo, an excellent solution would be to describe the methodology of performing gene sonodelivery. The title of the subsection "Intramuscular gene delivery" is too general, in fact it describes skeletal muscles as a target for sonodelivery of plasmid DNA.

Author Response

We appreciate the reviewer's time and acknowledgment of the importance of sonoporation for intramuscular gene delivery. We have made some modifications to the review article that we hope will serve to clarify points of concern brought up by the reviewer.

We feel that the brief presentation of other methods of gene delivery is critical in that it helps to provide background on the need for sonoporation, given that sonoporation helps to ameliorate many of the problems associated with these other techniques. A line has been added to the article (line 46-47) to clarify this point.

Because protocols for sonoporation have been established, and because the focus of this review is intended to be on presenting the benefits of sonoporation and compiling many of the different conditions used (sonoporators, settings, etc.) and subsequent results obtained, we feel that providing the methodology would both divert from the intended subject matter and replicate past literature. We have included a brief statement to this effect. We have also included a couple of new references and highlighted some of the current references that contain sonoporation protocols. Please see the new paragraph under section 3 (lines 141-145).

We have also adjusted some of the section headers and the title of the article to better reflect the more specific nature of the article (i.e. sonoporation in skeletal muscle).

Reviewer 2 Report

This is an excellent review of the use of ultrasound to facilitate gene delivery in the muscle.  It will be a valuable resource for other investigators. No edits needed.

Author Response

We appreciate the time and kind words of the reviewer and look forward to sharing our work with the research community!

Round 2

Reviewer 1 Report

The revisions made are adequate.